# A Comprehensive Self-Resistance Gene Database for Natural-Product Discovery with an Application to Marine Bacterial Genome Mining

**DOI:** 10.3390/ijms241512446

**Published:** 2023-08-04

**Authors:** Hua Dong, Dengming Ming

**Affiliations:** College of Biotechnology and Pharmaceutical Engineering, Nanjing Tech University, 30 South Puzhu Road, Jiangbei New District, Nanjing 211816, China

**Keywords:** resistance gene, RGDB, BGCs, marine microorganism, natural product

## Abstract

In the world of microorganisms, the biosynthesis of natural products in secondary metabolism and the self-resistance of the host always occur together and complement each other. Identifying resistance genes from biosynthetic gene clusters (BGCs) helps us understand the self-defense mechanism and predict the biological activity of natural products synthesized by microorganisms. However, a comprehensive database of resistance genes is still lacking, which hinders natural product annotation studies in large-scale genome mining. In this study, we compiled a resistance gene database (RGDB) by scanning the four available databases: CARD, MIBiG, NCBIAMR, and UniProt. Every resistance gene in the database was annotated with resistance mechanisms and possibly involved chemical compounds, using manual annotation and transformation from the resource databases. The RGDB was applied to analyze resistance genes in 7432 BGCs in 1390 genomes from a marine microbiome project. Our calculation showed that the RGDB successfully identified resistance genes for more than half of the BGCs, suggesting that the database helps prioritize BGCs that produce biologically active natural products.

## 1. Introduction

Natural products are secondary metabolites produced by organisms such as microbes and plants. They are usually small molecules not essential for their primary metabolism but often play important roles in ecological interactions and defense against predators or competitors. For example, many bacteria produce secondary metabolites with antibiotic properties that help them to compete for resources in their environment [1]. Almost a hundred years ago, the active-guide discovery of penicillin by Alexander Fleming’s serendipitous isolation of a new natural product called penicillin, produced by Staphylococcus colonies, occurred, which was proved to control bacterial growth by inhibiting cell division [2,3]. The tremendous clinical success of penicillin fueled the search for new chemicals from microorganisms, plants, and other natural resources over the past century, which led to the discovery of a large number of biologically active natural products, including β-lactam, aminoglycoside, tetracycline, macrolide, glycopeptide, lipopeptide, and other essential antibiotic skeletons and agrochemicals [4,5,6]. In the activity-guided approach to natural product discovery, highly efficient methods for activity screening, enrichment, and chemical isolation have been developed over the years to isolate pure bioactive compounds. Still, their discovery rate lags far behind the demand [7,8,9].

The emergence of next-generation sequencing (NGS) technology profoundly changed the traditional method of natural product discovery to the genome mining approaches [10,11,12]. The results of high-throughput sequencing have created a wealth of biological genomic data. For example, data analysis has shown that genes involved in natural product biosynthesis are closely aligned in the genome, forming so-called biosynthetic gene clusters (BGCs) [11,13,14]. In addition to genes encoding enzymes for the synthesis and modification of product backbone, metabolite transporters, and genes involved in the regulation of BGC expression, BGCs typically carry one or more genes, called self-resistance genes, encoding proteins that confer resistance to the natural product [15,16]. The basic logic behind self-resistance genes is that an organism must develop a mechanism to defend itself against possible damage caused by its metabolites, which, due to their biological activity, act on other organisms around it, thus ensuring that the organism gains an advantage over the competition [17]. This remarkable feature of BGCs has been used to develop the so-called Targeted Genome Mining (TDGM) methods, which first identify self-resistance genes and associated BGCs in the genome, and then predict the mode of action of natural products based on the mechanism of action of the resistance genes [16,18].

Another motivation for developing self-resistance gene-directed genome mining comes from the request to find novel antibiotic drugs in the fight against the antibiotic resistance crisis in which global public health is polluted by the widespread misuse of antibiotics [19,20]. Indeed, the lack of novel antibiotics with new modes of action makes finding new compounds to combat drug-resistant pathogens a critical task in the current drug development process [21]. On the other hand, long before humans began the mass production of antibiotics to prevent and treat infectious diseases, bacterial species had evolved self-resistant genes to tolerate antibiotics, thus allowing them to survive in crowded and harmful environments. Drug-resistant genes result from sanitation pollution from the massive improper use of antibiotics and provide a direction for discovering new antibiotics. Therefore, detecting resistance genes from the genome or metagenome becomes a critical step in identifying BGCs of bioactive products. This requires first compiling a knowledge database of all known antibiotic resistance genes [22], including sequence, structure, function, site of action, antibiotic compounds, and resistance mechanisms, and then using this database to mine genomes for possible resistance genes. So far, there are a variety of resistance gene databases available. One example of a resistance gene database is the Comprehensive Antibiotic Resistance Database (CARD) [23], which provides a centralized resource for researchers to access and analyze data on antimicrobial resistance genes. The CARD database includes over 2000 resistance genes, associated mutations, mobile genetic elements, and other relevant data. This database has been used to identify new resistance mechanisms and to track the spread of resistant organisms across different environments. The National Center for Biotechnology Information Bacterial Antimicrobial Resistance Gene Database (NCBIAMR) [24] contains annotated sequence records of representative DNA sequences encoding proteins that confer or contribute to resistance to various antibiotics. This dataset was constructed by aggregating collections from multiple sources, managing conflicts, and adding additional sequences discovered through a literature review, culminating in the integration of a selected database of reference genes and a selected collection of Hidden Markov Model (HMM) models [25]. Minimum Information about a Biosynthetic Gene Cluster (MIBiG) [26] is a biosynthetic gene cluster database for the management and storage of known BGCs, providing a powerful annotation standard for the annotation and metadata of BGCs and their molecular products. It facilitates the standardized storage and retrieval of BGC data and the development of comprehensive comparative analysis tools. It advances the next generation of research on the biosynthesis, chemistry, and ecology of a broad class of socially relevant bioactive secondary metabolites, guided by robust experimental evidence and rich metadata components.

Other databases include LacED [27], ARG-ANNOT [28], Resfam [29], DeepARG [30], ResFinder [31], ARTS-DB [18], etc., which, together, provide the basis for the TDGM discovery of natural products based on resistance gene screening.

A typical TDGM starts with genome annotation to identify all genes present in the genome, and then, on this basis, different methods are developed to predict putative BGCs. The most popular ones are based on homology comparisons, such as antiSMASH [32] and PRISM [33], and rely on identifying BGCs known in other microorganisms and detecting conserved domains and motifs associated with biosynthetic enzymes in these BGCs in the target genome. ClusterFinder uses a probabilistic model to predict the probability that a gene cluster is a BGC, based on the presence of known biosynthetic genes and their synteny with other genes [34]. Recently, there has been increasing interest in applying deep learning techniques, such as convolutional neural networks (CNNs) and recurrent neural networks (RNNs), to BGC identification by learning complex protein features and patterns from large-scale sequence datasets, which has been shown to outperform traditional machine learning methods. Tools that use this technique include, for example, DeepBGC [35], e-DeepBGC [36], iPRESTO [37], and others. These tools use a variety of algorithms to detect genes involved in secondary metabolisms, such as polyketide synthases (PKSs), nonribosomal peptide synthetases (NRPSs), hybrid PKS-NRPSs, and terpenoids. Another useful concept is the gene cluster family (GCF), which has emerged as a method to analyze BGCs on a large scale [34,38,39]. The GCF yields a network structure that significantly reduces the complexity of the BGC dataset and enables automatic annotation based on experimentally characterized reference BGCs. GCF networks have been used for the global analysis of bacterial biosynthetic spaces [34] and for bacterial genome mining at the scale of >10,000 genomes [40], and they have been integrated with metabolomics datasets for large-scale compound and BGC discovery [38]. In all these approaches, a notoriously tricky problem is predicting the natural product’s function from a BGC, which usually depends heavily on applying the knowledge of the known BGC and well-characterized resistance genes, such as MIBiG [26], ARTS-DB [18], etc. Therefore, in addition to the functional analysis of necessary genes, the resistance genes must be identified and compared with a comprehensive dataset to assess the function novelty of the product and thus prioritize BGCs.

In this study, we first compiled a new resistance gene database (RGDB) by integrating existing resources and applying additional detailed annotation, which clarified their intrinsic relationship with the natural production encoded in BGCs. The database and script codes for predicting resistance genes and their activities using the database are provided at https://github.com/mingdengming/rga (accessed on 23 February 2023). As an application, the method was used to analyze the MarRef bacterial genome dataset, which consists of more than one thousand genomes from the Marine Metagenomic Portal (MMP) [41]. As a genomic dataset for marine bacteria, the recently disclosed MarRef dataset has a good degree of completeness and systematicity and thus allows for the systematic detection of the occurrence of resistance genes in the RGDB in a given system. Both the resistance genes and some of the encoded natural products of the BGCs were predicted and analyzed using the database. The RGDB established in this work might enlarge our knowledge about resistance genes and mechanisms; it may help annotate the function of BGCs and facilitate TDGM in large-scale natural production mining.

## 2. Results and Discussion

### 2.1. The Resistance-Gene Database

Table 1 shows the number of resistance genes collected from four major data repositories. Among these four databases, the resistance genes collected from CARD have detailed annotation information. Therefore, they are directly used in our database without the need for further modification or additional annotation. MIBiG records valuable BGC data, including well-characterized product information [26]. The mechanisms of most resistance genes were determined manually from annotations in the database and subsequently validated by comparison with the CARD database. A total of 629 hidden Markov model profiles of resistance genes were collected from NCBIAMR, and each of them was manually annotated with their involved resistance mechanism and compound. The functional annotation of the resistance genes found in the Uniprot database is less than that given by the other three databases. Except in the case of the efflux pump, most of the annotation is completed by comparing these genes with resistance genes recorded in the other three libraries by transferring the annotation obtained on the linkage to the query Uniprot genes. Table 1 also provides statistics on the annotated resistance mechanisms in the database, showing that product inactivation is the most frequent among the five resistance mechanisms, followed by the efflux pump, while the target modification mechanism is the least frequent. Appendix A lists all 7297 entries of resistance gene sequences and 629 HMM models of resistance gene profiles in the database.

As a demonstration of the application of the RGDB database, it is interesting to study the distribution of resistance genes inside the recently disclosed marine bacterial genomes and, in particular, to analyze their relationship with the natural products that may be synthesized by the gene clusters of these bacteria.

### 2.2. The BGCs in the Bacterial Genomes of MarRef

The marine microbial genome is considered one of the richest resources for natural products due to the diverse environmental conditions in which marine microorganisms live; the genome has proven extremely valuable in discovering new natural products with different structures and biological activities. An important reason for the diversity of marine natural products may be that microbial populations compete to win by synthesizing complex, biologically active, and diverse compounds to inhibit the growth of other similar populations in survival competition. We applied the RGDB to analyze a recently published marine microbial genome dataset, i.e., the MarRef dataset. A total of 1403 genomic sequences were downloaded from GenBank [42] for the MarRef dataset, 13 of which could not be processed with antiSMASH 6.0 due to formatting errors, resulting in a total of 1390 genomes being analyzed in this study. The detailed species taxonomy of the selected organism is listed in Appendix A, which is summarized in Figure 1A. Parsing these genomes with antiSMASH yielded 7432 BGCs, ranging from 5 to 670 kb. Figure 1B shows the distribution of the broad classes of BGC products detected, with the most abundant being RiPPs (22%), followed by terpenoids (14%) and NRPS (12%). Note that antiSMASH could not even determine, for the remaining 40% of the BGC products, which class of compounds they belonged to, let alone their specific compound structures and activities. Statistical analysis by species origin of BGCs showed that the three major bacterial producers of secondary metabolites are *Proteobacteria*, *Actinobacteria*, and *Firmicutes*, which together contributed more than 80% of the BGCs, and the archaeal producer *Euryarchaeota*, which contributed 2% of the BGCs (Appendix A). BGCs from other clades contribute much less and are therefore not shown. As previously observed, the number of BGCs per genome varies considerably among bacteria.

### 2.3. Preliminary Product Annotation of BGCs by Organizing Gene Clusters into Gene Cluster Families

Identifying BGCs with known metabolites is essential to genome mining, allowing researchers to prioritize known and unknown biosynthetic pathways for new natural product discovery [43]. Gene cluster family (GCF) analysis enables the large-scale annotation of query BGCs based on their similarity to the reference BGCs in MIBiG with well-characterized product and derivative information. After GCF analysis, these marine BGCs were first grouped into families using BiG-SCAPE, yielding the gene cluster families (GCFs). BiG-SCAPE [39] analysis showed that the studied BGCs were mainly clustered together within orders, resulting in a set of 5803 GCFs (of which, only 518 families had more than three members) organized into a network based on their genetic components and arraignment within the reference BGCs (Figure 2). The analysis resulted in 5803 GCFs, of which, only 208 GCFs, i.e., 3.6% of the total, had a corresponding 327 reference BGCs in their family. The product information of these GCFs was then predicted based on that of the reference BGCs, resulting in 23 RiPPs, 10 terpenes, 94 NRPS, 102 PKS, 31 PKS-NRPS, and 67 other products. Note that hybrid BGCs, such as those of PKS-NRPS, might be presented in more than one family; a similar scenario was found in previous GCF analyses [38].

### 2.4. Identifying Resistance Genes in MarRef BGCs Using RGDB

The identification of resistance genes in MarRef BGCs is straightforward using the newly constructed RGDB, which gives 7079 putative resistance genes distributed in 3878 BGCs (see Appendix A for details; also see http://www.mingbioinfo.online/rga/marrefdata (accessed on 23 February 2023)). These identified resistance genes were classified according to their resistance mechanisms, the most common being efflux pumps, with a total of 4183. Most efflux pumps are ATP-binding cassette (ABC) transporters and major facilitator superfamily (MFS) transporters. Some efflux pumps have narrow substrate specificity, such as tetracycline efflux, while others are relatively broad. As the most typical mechanisms of product inactivation, aminoglycoside acetyltransferase, aminoglycoside phosphotransferase, and glyoxylate/bleomycin resistance proteins repeatedly appear in these MarRef BGCs. Vancomycin resistance genes were found to be the most frequently occurring genes for target-related mechanisms. It is worth mentioning that the 3-oxoacyl-[acyl-carrier-protein] synthase used for fatty acid biosynthesis is present in many BGCs. Previous studies have found that copy number can be increased by self-replication and mutation, thus providing resistance to the host [44,45]. The analysis of these putative resistance genes showed that 118 of these 3878 BGCs detected to have resistance genes did carry multiple resistance genes of different types and mechanisms, without considering the efflux pump mechanism (see the two cases in Table 2). Thus, resistance mechanisms may not be unique for each bacterium, and multiple resistance mechanisms may evolve depending on the compound, which supports previous findings.

### 2.5. The Self-Resistance Gene-Directed Natural-Product Discovery

As an application of the RGDB database, self-resistance gene-directed screening may lead to the identification of BGCs from a vast number of microbial genomes, whose encoded natural products might have potentially higher biological activity due to the resistance. In the Marref genome database, BGCs containing self-resistant genes were first collected through RGDB screening and then analyzed via the GCF [34] method to determine the type of products encoded. Specifically, if a GCF containing a MarRef BGC also includes known BGCs from the MIBiG database, then the known compounds encoded by MIBiG BGCs can be used to predict the types of the natural product encoded by the MarRef BGC, including the chemical scaffolds and their possible interactions with the involved self-resistance genes. The annotation of the resistance mechanisms provides insights into the biological activities of the encoded natural products. Appendix A records 51 GCFs that were screened using this method, including 164 MarRef BGCs and encoded natural product frameworks.

Table 3 lists three typical examples from the self-resistance gene-directed natural-product discoveries. As the first example, the MarRef BGC was detected with a resistance gene tetB(P), which encodes a tetracycline ribosome protection protein. This gene cluster was assigned to the GCF_01148 family and listed along with the known MIBiG BGC0000853, identified to synthesize the compound ectoine. Therefore, based on the GCF evolutionary relationship, it is reasonable to hypothesize that the product of CP018047.1.region026 may have a chemical scaffold similar to ectoine, which is a potential inhibitor to protein synthesis, thereby exhibiting antibacterial bioactivity. The second example involves five MarRef gene clusters, all containing the ABC-F resistance gene and belonging to the GCF_05426 family. Through the MIBiG gene cluster BGC0001502 in this GCF, it is inferred that these BGCs synthesize an NRP-type compound called amonabactin P 750. The evolutionary analysis thus suggests that these MarRef BGCs can produce similar amonabactin P 750, which should have inhibitory effects on ribosomal protein synthesis. In the final example, the MarRef BGC was assigned to the GCF_07814 family that contains the typical resistance gene aac(3), which encodes an aminoglycoside acetyltransferase. The evolutionary analysis indicates that the product scaffold of CP043317.1.region037 is similar to lobophorin A/B, a macrocyclic lactone with antibacterial bioactivity. The bacterium can acquire resistance to this natural product by modifying the transferred aminoglycoside acetyl group.

### 2.6. Analysis of Resistance Genes in Terpene-Producing BGCs

As a diverse class of natural products with a wide range of biological activities produced by microorganisms, it is of great interest to study the coding of terpenoids in marine microbial BGCs and the role they may play in self-resistance. Of the 3878 BGCs screened that contained putative resistance genes, 553 were terpene-producing BGCs. Table 4 shows the selected 4 typical terpenoid BGCs for which resistance genes and products were identified, and details of the 553 BGCs are also listed in Appendix A. Figure 3 shows the location of the resistance genes in the gene cluster for these four BGCs, indicating that the location of the resistance genes could occur anywhere in the BGC map. Figure 4 compares the structure of resistance gene proteins with their corresponding reference resistance gene proteins. The RMSD values calculated based on structural comparison ranged from 0.1 to 2 Å, indicating that their structural similarity is very high, verifying that the queried resistance protein may be functionally identical to the reference protein, and the functional annotation of the corresponding resistance gene should also be reliable.

## 3. Materials and Methods

### 3.1. Typical Resistance Mechanisms Used to Characterize Resistance Genes

Five typical microbial self-protection mechanisms were identified based on the biological relationship between resistance genes and the natural products: efflux pump, product inactivation, target protection, target alteration, and target replacement.

Efflux pumps are specialized transport proteins that can pump antibacterial or anticancer drugs out of cells, thereby reducing their effectiveness [46]. They are among the highest percentage of proteins encoded by resistance genes. In natural product modification, resistance genes encode enzymes able to add chemical entities (e.g., phosphates or acetates) to the small molecule structure of the natural product, causing structural changes that affect the binding to the targets, thus producing self-resistance to the products [47,48]. Natural products often contain chemical bonds such as amide and ester bonds that are susceptible to breakdown and whose integrity is critical for biological activity. Some enzymes in organisms can target the cleavage of such bonds, providing natural product degradation, a means of self-resistance for the organism [49]. As a typical example, bacteria can develop resistance to the antibiotic penicillin by producing an enzyme called beta-lactamase, which breaks down the penicillin molecule and renders it ineffective [50]. In this work, product modification and product degradation are regarded as types of product inactivation. Target protection employs a resistance protein that physically interacts with an antibiotic target to rescue it from the antibiotic-mediated inhibition [51]. Target alteration involves mutating and modifying the target proteins of action of the natural-product small molecule, thereby interfering with the binding of the small molecule to the target and thereby conferring resistance [52]. Examples include point mutations, deletions, and insertions in RNA polymerase [53] and those in DNA gyrase mutations [54], resulting in resistance to rifamycins and quinolones, respectively. Recently, a target replacement mechanism has been identified in which mutant proteins encoded by resistance genes are homologous to essential wild-type proteins targeted by the BGC products and retain the original function of the target proteins. The mutations render the self-resistant enzyme insensitive to natural products [55]. In general, there is a high probability that target replacement is generated by point mutations in specific genes (housekeeping enzymes), resulting in relatively rapid and simple resistance for the producer with minimal impact on the producer’s [16]. In building the database, resistance genes were annotated in the above-mentioned mechanistic categories based on records from databases such as MIBiG, CARD, and NCBIAMR and through literature searches, thus providing a reference for understanding the function of biosynthesis gene clusters containing these resistance genes.

### 3.2. Preparation of Sequence and Mechanism Annotations of Resistance Genes

Most of the resistance genes were collected from three existing databases, namely the Minimum Information on Biosynthetic Gene Clusters (MIBiG) [26], the Comprehensive Antibiotic Resistance Database (CARD) [56], and the National Center for Biotechnology Information Bacterial Antibiotic Resistance Reference Gene Database (NCBIAMR) [24]. The sequences of resistance genes were extracted from the CARD database version 3.2.2. Special attention was paid to collecting relevant mechanisms of action of each resistance gene and recording the acting antibiotic compounds for further characterization. The third part of the resistance gene data was obtained from the resistance protein homology model extracted from the NCBIAMR database. The involved chemical compounds and resistance mechanisms acted by this part of the resistance genes were annotated according to the “DESC” records in the database. The proteins encoded by BGCs in MIBiG version 2.0 were screened to identify resistance genes based on annotations using the keyword “resistance.” The associated natural products of BGCs were also identified as possible target compounds for the action of resistance genes. Resistance mechanisms were identified by first checking the annotation of the genes in MIBiG and, if not in MIBiG, the annotation of homologous genes found in CARD and NCBIAMR.

The resistance genes recorded in the UniProt database were also screened against “resistant” and “resistance” keywords. To avoid duplication with the existing resistance genes, these raw data from UniProt were de-duplicated by comparison with the dataset already obtained above. Specifically, these resistance genes were compared with the resistance gene data found in CARD and MIBig by applying for the BLAST program [57] and were removed if the resulting e-values were not greater than 1 × 10^−20^ and the bit scores were no less than 50. They were also compared with NCBIAMR resistance modules using the HMMSCAN program [25], and those with an e-value smaller than 1 × 10^−16^ and a bit-score greater than 50 were removed. Although genes containing resistance keywords may be better curated in UniProt than in CARD and NCBIAMR, if UniProt did not provide relevant resistance compounds and resistance mechanisms, these data were alternatively obtained based on a homology search with the datasets generated based on CARD and NCBIAMR described above.

### 3.3. Compiling the Resistance Gene Database and Preparing the Knowledge-Based Annotation of BGCs

The accurate prediction of natural products synthesized by gene clusters and their biological activities remains a challenging topic in bioinformatics; in particular, there is still no reliable method to predict the biological activities of gene clusters encoding terpenoids [58]. In this work, we apply knowledge of the self-resistance mechanisms adopted by resistance genes to infer the biological activity of BGCs in identifying the resistance genes carried by the gene clusters. To this end, we first compiled the datasets prepared in the above section into a resistance gene database (RGDB). The database consists of two parts: a part derived from CARD, MIBig, and Uniprot, stored in FASTA format and queried by BLAST search, and a second part derived from NCBIAMR, a Hidden Markov database, searched with the hmmscan program. Each resistance gene sequence or gene model in the database is pre-annotated with information on the resistance mechanism, particularly the compounds that may be involved, according to their origins. Specifically, resistance genes from CARD may be annotated with information on antibiotic compounds, resistance genes from MIBig may be annotated with information on natural product molecules, resistance genes from UniProt may be annotated with information on enzyme products, and resistance models from NCBIAMR may be annotated with information on antibiotic molecules based on the DESC statement. These compounds, if applicable, provide valuable information for predicting the biological activity of the inquired resistance genes and the products of the inquired BGCs. In this work, an in-house script code is provided, which can automatically scan the query sequence in the RGDB and return the mechanism and chemical compound annotation of matched resistance genes in the successful hit, as the resistance mechanism prediction of the query.

### 3.4. The RGDB-Based Analysis of a Marine Microorganism Genome Dataset

Marine microorganisms are important sources of natural products since they have evolved in an incredibly diverse and extreme environment, leading to specialized metabolites. Here, we applied the RGDB-based method to analyze the diversity of a marine microbial genomic dataset, MarRef [41], a manually curated reference genomic database of marine microorganisms, where each genome was fully sequenced. Each entry in MarRef includes 120 metadata fields, including information about the sampling environment or host, organism and taxonomy, phenotype, pathogenicity, assembly, and annotation. 1403 genomes were downloaded from the MMP database (https://mmp2.sfb.uit.no/marref/ (accessed on 13 December 2022)) as of MarRef version 1.7. Each genome was first analyzed using antiSMASH 6.1.0 with default parameters [32] to resolve BGCs. Next, the BiG-SCAPE [39] version 1.1.4 was used to construct a BGC sequence similarity network to analyze gene cluster diversity associated with enzyme phylogeny by constructing BGC family GCFs. The networks were built using a similarity threshold of 0.3, as higher thresholds resulted in a wide range of large proposed BGC families [59].

Identifying resistance genes in MarRef-BGCs based on RGDB can be straightforward by using the script mentioned above. Specifically, for a given MarRef-BGC, each protein recorded in the coding sequence (CDS) will be matched against the RGDB. A hit is a success only if the alignment E-value is less than the cutoff value 1 × 10^−30^. Once a successful hit is made, the script automatically assigns a resistance annotation with the resistance mechanism and compound, which may be used further to annotate the biological activity of the BGC under study.

### 3.5. Structural Annotation of Selected Resistance Genes of Terpenoid BGCs

Terpenoids, as one of the most diverse and widely distributed natural products, have been a hot research topic in microbial natural products. One of the critical issues in this study is understanding the function and structure relationships between the terpenoid products and the terpene synthases in the gene clusters. Here, the RGDB-based method was applied to the MarRef-BGCs to identify those terpenoid BGCs containing resistance genes. Specifically, we randomly selected one of four types of resistance genes: product inactivation, target alteration, target protection, or target replacement. Due to their widespread presence in microbial BGCs and the absence of corresponding specific compounds, efflux pump resistance genes have yet to be studied, and further research is needed. The structures of the proteins encoded by the four resistance genes obtained by RosettaFold [60] were compared with those of the corresponding resistance genes in the RGDB. The structures of the resistance genes in the RGDB were generated by alphafold [61] and obtained through an access link provided by Uniprot.

## 4. Conclusions

This study presented a comprehensive resistance gene database (RGDB) by collecting data from four major databases: CARD, MIBIG, NCBIAMR, and UniProt. The database was designed for the analysis of the self-resistance genes in BGCs in target-driven genome mining. Both possible resistance mechanisms and compounds involved in the bioactivities of the resistance gene in the database were annotated. Scripts were provided for scanning the database to identify resistance genes in query BGCS and the relevant compounds involving the resistance mechanisms. We applied RGDB-based annotation to analyze resistance genes in 7432 BGCs in 1390 genomes in the MarRef database and found 7079 resistance genes in 3878 BGCs, of which, 59% are efflux pumps. Excluding these pumps, 3% of these resistance-gene carrier BGCs were found to have more than one type of resistance gene and mechanism. Notably, most of the resistance genes in the database were prepared using terrestrial microorganisms. However, it could still identify resistance gene candidates for more than half of all the marine microbial BGCs studied. All five types of resistance mechanisms were found in marine microorganisms, and the associated compound of the resistance genes provides the starting point for inferring the possible products of BGCs. We also note that nearly half of the tested marine BGCs have no resistance genes identified by the current RGDB. This suggests that unique self-protection mechanisms may exist in these marine microorganism BGCs that need to be discovered in future studies using new resources.

The resistance genes in the database are those that already exist. Regarding the fact that organisms follow the environment to evolve new resistance genes, they may not be detected, but a preliminary judgment can be made based on the similarity of gene sequences. In the process of co-evolution of resistance genes with the host, the original ones may be invalid, and the host will rapidly change its genome composition through mutation or recombination to ensure its adaptation to the current environment, so the resistance genes detected according to this database may not be the current effective resistance genes. Therefore, the database of resistance genes should be able to keep up with the real-time research information and make timely supplemental adjustments to the annotation information to ensure the usability of the database annotation results. When we carry out the prediction of genome mining orientation, we can annotate the possible resistance genes in BGC, but whether we can give full play to the role of orientation may not be realized only by the database; only the target-related resistance mechanism can predict the target and biological activity of the product.

## Figures and Tables

**Figure 1 ijms-24-12446-f001:**
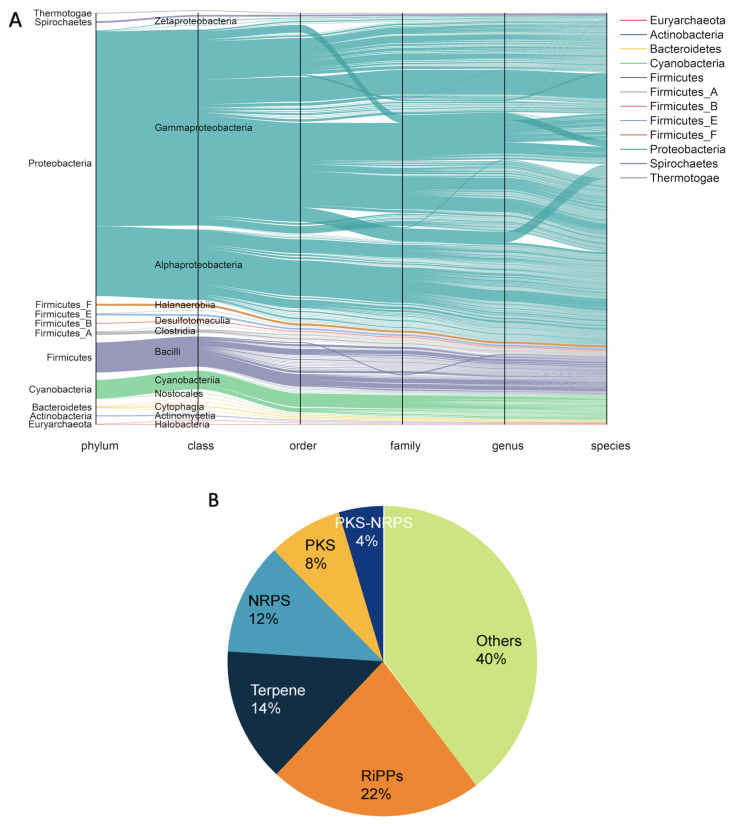
(**A**) the microbial genome sequences deposited in the MarRef database, (**B**) the distribution of MarRef BGCs according to their products.

**Figure 2 ijms-24-12446-f002:**
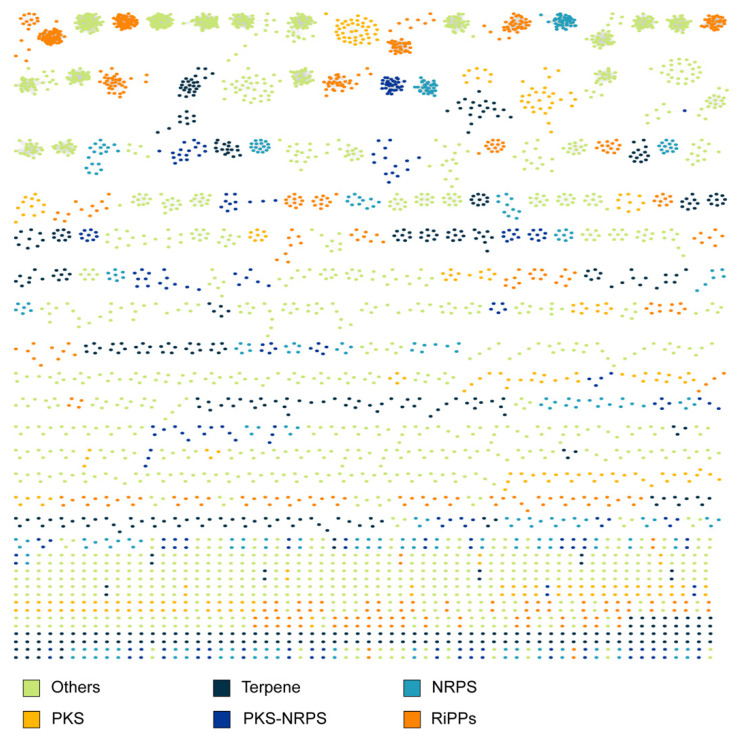
Similarity network of known and query BGCs. Known BGCs are from the MIBiG v2.0 database; the queried BGCs are from the MarRef bacterial genomes.

**Figure 3 ijms-24-12446-f003:**
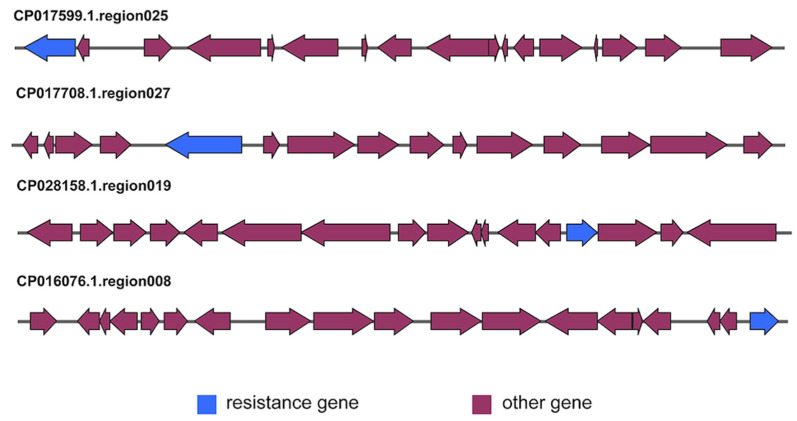
Comparison of gene sequence maps of four selected terpenoid MarRef-BGCs. The blue block represents the predicted resistance genes in the gene sequence map of the BGCs, and the red ones represent other functional genes.

**Figure 4 ijms-24-12446-f004:**
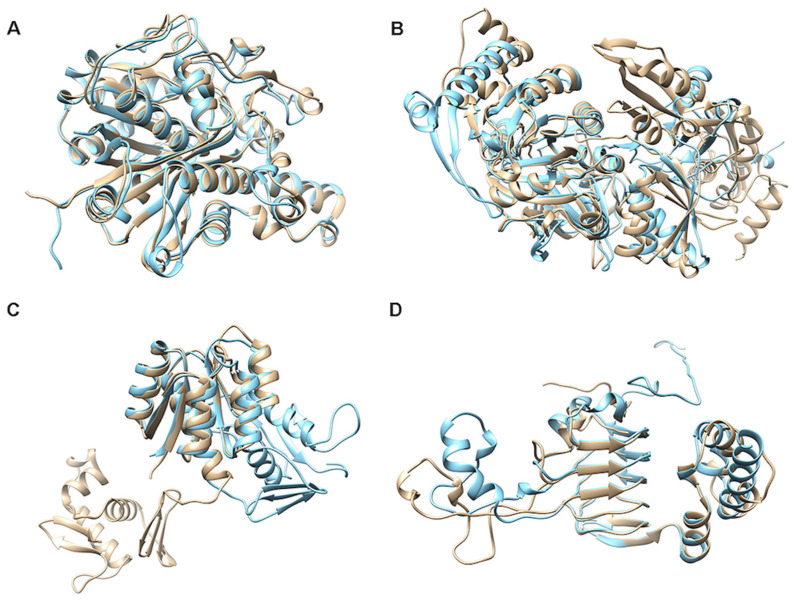
The structural alignment of proteins encoded in the resistance genes of the four selected terpenoid MarRef-BGCs with those of their reference resistance genes found in RGDB. (**A**) CP0175990001 (blue) vs. ACS13710.1 (yellow), a protein encoded by a resistance gene found in MIBiG BGC0001140.1, (**B**) CP0177080005 (blue) vs. AAA20117.1 (yellow), a tetB(P) protein with tetracycline antibiotic resistanc encoded by the CARD resistance gene ARO:3000195, (**C**) CP0281580014 (blue) vs. AAA65953.1 (yellow), a protein involving resistance to penicillin-like antibiotics encoded by the CARD resistance gene ARO:3002919, (**D**) CP0160760019 (blue) vs. AAA22081.1 (yellow), a protein with phenicol antibiotic resistance encoded by the CARD resistance gene ARO:3004451.

**Table 1 ijms-24-12446-t001:** The statistics of resistance genes and resistance mechanisms of the RGDB.

Resource	Efflux Pump	Product Inactivation	Target Alteration	Target Replacement	Target Protection	Total	Data Type
MIBiG ^(a)^	79	35	9	1	2	126	FASTA
CARD ^(b)^	259	3891	209	67	147	4573	FASTA
NCBIAMR ^(c)^	72	396	91	26	44	629	HMM
UniProt ^(d)^	1146	355	1043	24	30	2598	FASTA

The database versions used are, (a) v2.0, (b) v3.2.2, (c) 2022.04.07, (d) Swiss-Prot, 30 October 2022.

**Table 2 ijms-24-12446-t002:** Examples of MarRef BGCs that contain two types of different resistance mechanisms.

MarRef_BGC_ID	Species	BGC_CDS_ID	RGDB_Subject	Resistance	Product Class
AP008957.1.region010	*Rhodococcus erythropolis* PR4	AP0089570009	AAA26793.1	efflux pump	macrolide antibiotic
		AP0089570023	AAA26684.1	target protection	lincosamide antibiotic
BA000028.3.region001	*Oceanobacillus iheyensis* HTE831	BA0000280007	AAB36568.1	product inactivation	phenicol antibiotic
		BA0000280017	AAA99504.1	efflux pump	peptide antibiotic

**Table 3 ijms-24-12446-t003:** Examples of the self-resistance gene-directed natural-product discovery.

GCF ID	MarRef_BGC	Species	BGC_CDS_ID	RGDB_Subject	Resistance	Product Class
GCF_01148	CP018047.1.region026	*Streptomyces niveus strain* SCSIO 3406	CP0180470001	AAA20117.1	target protection	ectoine
GCF_05426	CP022181.1.region002	*Aeromonas salmonicida* strain S44	CP0221810025	ribo_prot_ABC_F-NCBIFAM	target protection	amonabactin P 750
GCF_07814	CP043317.1.region037	*Streptomyces olivaceus* strain SCSIO T05	CP0433170057	ARO:3002539	product inactivation	lobophorin B/A

**Table 4 ijms-24-12446-t004:** The resistance genes of four terpenoids BGCs and their corresponding resistance-relevant product class.

Terpenoid BGCs	Species	BGC_CDS_ID	RGDB_Subject	Resistance	Product Class
CP017599.1.region025	*Moorena producens* PAL-8-15-08-1	CP0175990001	ACS13710.1	target replacement	diterpenoid antibiotic
CP017708.1.region027	*Moorena producens* JHB	CP0177080005	AAA20117.1	target protection	tetracycline antibiotic
CP028158.1.region019	*Plantactinospora* sp. BC1	CP0281580014	AAA65953.1	target alteration	glycopeptide antibiotic
CP016076.1.region008	*Actinoalloteichus fjordicus*	CP0160760019	AAA22081.1	production inactivation	phenicol antibiotic

## Data Availability

Both the database and the scanning and analysis codes are available from https://github.com/mingdengming/rga (accessed on 23 February 2023).

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
