# Peer review of "A Comprehensive Self-Resistance Gene Database for Natural-Product Discovery with an Application to Marine Bacterial Genome Mining"

_ijms, 2023, doi:10.3390/ijms241512446_

Round 1
Reviewer 1 Report
In the presented manuscript, Hua Dong and Dengming Ming presented “A comprehensive self-resistance gene database for natural product discovery with an application to marine bacterial genome mining”.
The manuscript is not suitable for publication in Antibiotics, the language should be revised and additional analyses should be performed to present a comprehensive analysis as proposed by the authors.
I provide some major and minor comments below.
Major;
1. The authors mention that they "compiled" and not that they created a new database. It should be clear in the manuscript what is the significant contribution of the new database presented because if they merely compile data from existing databases, the contribution is not significant. In addition, they mention at the end of the introduction "it may help annotate the function of BGCs and facilitate TDGM in large-scale natural production mining".
They should indicate, "it will help annotate the function of BGCs".
2. Figure 1 is not very informative since it only contains the phyla, the authors should present a graph with more information. They can use a Sankey chart and include the taxonomic categories: Phylum > Class> Order> Family> Genus> Species. This graph will give a better insight to the study.
3. Supplementary files cannot be opened
Minor,
The writing and grammar of the article must be revised.
1. Line 76-78. Contains smaller font size than the rest of the text
2. Line 77. Remove the "etc" and add the examples you summarized in the "etc".
Moderate editing of English language required
Reviewer 2 Report
Introduction: In this study 3 databases were utilized (MIBiG, 163 CARD, and NCBIAMR) to build RGDB, only CARD was mentioned in introduction. It would be better if you mention the other 2 databases.
Line 75-79 - Please keep the same font formatting.
Line 122: It will be easy for the readers to understand why you use this MarRef bacterial genome dataset.
Conclusions: are there any limitations for analysis of the self-resistance genes in BGCs in target-driven genome mining using RGDB mining. If there are any limitations please include.
The manuscript was exceptionally well written, demonstrating a strong command of English and a clear understanding of the subject matter. The language was precise, concise, and engaging, making it a pleasure to read. I would like to highlight that I only came across one minor formatting error while reviewing the manuscript. This indicates your careful attention to detail and overall professionalism in preparing the document. I recommend rectifying the formatting issue, which will further enhance the visual appeal of your manuscript.
Round 2
Reviewer 1 Report
The authors have responded to the comments, so the manuscript is suitable for publication.
Minor editing of English language required